# High Correlation among Brain-Derived Major Protein Levels in Cerebrospinal Fluid: Implication for Amyloid-Beta and Tau Protein Changes in Alzheimer’s Disease

**DOI:** 10.3390/metabo12040355

**Published:** 2022-04-15

**Authors:** Kyoka Hoshi, Mayumi Kanno, Mitsunari Abe, Takenobu Murakami, Yoshikazu Ugawa, Aya Goto, Takashi Honda, Takashi Saito, Takaomi C. Saido, Yoshiki Yamaguchi, Masakazu Miyajima, Katsutoshi Furukawa, Hiroyuki Arai, Yasuhiro Hashimoto

**Affiliations:** 1Department of Biochemistry, Fukushima Medical University, Fukushima 960-1295, Japan; khoshi@fmu.ac.jp; 2Department of Forensic Medicine, Fukushima Medical University, Fukushima 960-1295, Japan; kannoma@fmu.ac.jp (M.K.); ponchan@fmu.ac.jp (T.H.); 3Department of Neurology, Fukushima Medical University, Fukushima 960-1295, Japan; mitzabe@gmail.com (M.A.); maaboubou@gmail.com (T.M.); ugawa@fmu.ac.jp (Y.U.); 4Center for Integrated Science and Humanities, Fukushima Medical University, Fukushima 960-1295, Japan; agoto@fmu.ac.jp; 5Laboratory of Proteolytic Neuroscience, RIKEN Center for Brain Science, Saitama 351-0198, Japan; saito-t@med.nagoya-cu.ac.jp (T.S.); takaomi.saido@riken.jp (T.C.S.); 6Structural Glyocobiology Team, RIKEN Global Research Cluster, Saitama 351-0198, Japan; yyoshiki@tohoku-mpu.ac.jp; 7Department of Neurosurgery, Juntendo University, Tokyo 113-8421, Japan; mmasaka@juntendo.ac.jp; 8Institute of Development, Aging and Cancer, Tohoku University, Miyagi 980-8575, Japan; katsfuru@hotmail.com (K.F.); hiroyuki.arai.b5@tohoku.ac.jp (H.A.)

**Keywords:** Alzheimer’s disease, neurodegenerative diseases, cerebrospinal fluid, lipocalin-type prostaglandin D2 synthase, transferrin

## Abstract

The cerebrospinal fluid (CSF) plays an important role in homeostasis of the brain. We previously demonstrated that major CSF proteins such as lipocalin-type prostaglandin D2 synthase (L-PGDS) and transferrin (Tf) that are biosynthesized in the brain could be biomarkers of altered CSF production. Here we report that the levels of these brain-derived CSF proteins correlated well with each other across various neurodegenerative diseases, including Alzheimer’s disease (AD). In addition, protein levels tended to be increased in the CSF samples of AD patients compared with the other diseases. Patients at memory clinics were classified into three categories, consisting of AD (*n* = 61), mild cognitive impairment (MCI) (*n* = 42), and cognitively normal (CN) (*n* = 23), with MMSE scores of 20.4 ± 4.2, 26.9 ± 1.7, and 29.0 ± 1.6, respectively. In each category, CSF protein levels were highly correlated with each other. In CN subjects, increased CSF protein levels correlated well with those of AD markers, including amyloid-β and tau protein, whereas in MCI and AD subjects, correlations declined with AD markers except p-tau. Future follow-up on each clinical subject may provide a clue that the CSF proteins would be AD-related biomarkers.

## 1. Introduction

Alzheimer’s disease (AD) is the most common cause of dementia [1], but the link between pathophysiological processes and the emergence of dementia has not been fully elucidated. Sperling et al. hypothesized that the AD pathology begins many years prior to the onset of dementia, and that amyloid-β (Aβ) peptide accumulation is a key event in the initial pathophysiology [2]. Aβ is produced by the sequential proteolytic cleavage of amyloid precursor protein (APP), a membrane-bound protein that is initially cleaved extracellularly by β-secretase, generating a soluble APP fragment and a stub (C99) in the membrane [3]. The C99 stub is further cleaved by β-secretase, generating extracellular Aβ and intracellular peptide fragments. In this process, several isoforms of Aβ, the most abundant of which is the 40-amino acid isoform (Aβ40), can be produced and subsequently diffuse to the CSF. Although a minor component, a second 42-amino acid isoform (Aβ42) is preferentially deposited in amyloid plaques due to its hydrophobic properties. As such, the Aβ42 isoform content in the CSF begins to decrease early in AD progression, making this trait a marker for AD diagnosis. The Aβ42/Aβ40 ratio shows better diagnostic accuracy than Aβ42 alone because the change in Aβ42 can be normalized to the major Aβ40 isoform [2]. Amyloid plaques can be visualized, even in living individuals, by amyloid positron emission tomography (PET) [4]. With this technique, the PET signal intensity and the cerebrocortical distribution of amyloid support the diagnosis of AD. In addition, the PET signal shows good concordance with CSF biomarkers such as Aβ42 and the Aβ42/Aβ40 ratio [5]. A longitudinal study revealed that some cognitively normal (CN) and mild cognitive impairment (MCI) subjects converted from PET-negative to PET-positive during follow-up [6]. Prior to follow-up, Aβ deposits may be insufficient to show PET positivity, suggesting that Aβ changes in CN and MCI subjects may take place over time prior to PET positivity. Thus, the initial pathophysiological changes in AD remain difficult to detect. In later stages of the disease, the formation of amyloid plaques is followed by the abnormal metabolism of tau. Tau is an intracellular microtubule-associated protein that is associated with axonal transport. In AD pathology, tau is hyperphosphorylated (p-tau) [2], detaches from the microtubules, and is deposited in intracellular neurofibrillary tangles. Leakage of intracellular tau and p-tau into the CSF is associated with neuronal injury and death, the progression of which is manifested in the form of dementia. As CSF tau and p-tau are also representative diagnostic markers for AD, they are collectively referred to, along with Aβ42, as “AD core biomarkers” [7].

The cerebrospinal fluid (CSF) plays a crucial role in homeostasis of nutrients and pH, etc. We previously demonstrated that major CSF proteins could serve as biomarkers of altered CSF production in spontaneous intracranial hypotension (SIH) and idiopathic normal pressure hydrocephalus (iNPH) [8,9,10,11]. Patients with SIH show cranial hypotension (<60 mm H_2_O) due to CSF leakage [10]. Myelography often reveals the location and extent of a CSF leak. Radioisotope (RI) scintigraphy reveals rapid disappearance of RI from subarachnoid space, providing quantitative data on CSF leakage. The leakage and flowing cranial hypotension were speculated to induce a compensatory increase of CSF production [10]. Concomitantly, major CSF proteins secreted from choroid plexus were increased; i.e., lipocalin-type prostaglandin D2 synthetase (L-PGDS) and brain-derived transferrin (Tf). In addition, these proteins were inversely correlated with levels of cranial hypotension and RI disappearance [11]. In contrast to SIH, these markers were decreased in the CSF of patients with iNPH, in which the production of CSF is thought to decrease [11,12,13]. INPH is treated by shunt operation, which bypasses excess CSF from ventricle to peritoneal cavity, leading to normalization of CSF production. Within 3 months after the shunt operation, the levels of brain-derived Tf increased to control levels [14], suggesting that the brain-derived Tf could be a marker of altered CSF production. Tf glycan-isoform analysis also reveals that different brain-derived Tf isoforms, but not blood-derived isoforms, could be markers of CSF production. CSF Tf is composed of three glycan isoforms: Tf with sialic acid-terminated glycans from blood (Sia-Tf), Tf with *N*-acetylglucosamine (GlcNAc)-terminated glycans, possibly from the CSF-producing choroid plexus [8,12,13], and Tf with mannose-terminated glycans from neurons (Man-Tf) [15]. Indeed, Man-Tf and GlcNAc-Tf are not detected in the CSF of congenital hydranencephaly patients, in which the cerebrum is mostly lacking [15]. Thus, brain-derived Tf levels could be markers of altered CSF production, but blood-derived Sia-Tf is unlikely to be such a marker.

In the present study we analyzed CSF levels of brain-derived Tf, L-PGDS, and AD markers in neurodegenerative diseases and examined their correlations.

## 2. Results

### 2.1. Major Proteins in Serum and Cerebrospinal Fluid

Blood serum contains albumin, immunoglobulins, and Tf as major proteins (Figure 1A). These proteins are also major components of the CSF, in addition to L-PGDS and TTR. SDS-PAGE analysis reveals that Tfs were detected around 70–73 kDa area as two bands, one of which migrates slightly slower than the other (Figure 1A). Both bands can be seen on immunoblots to react anti-Tf antibody (Figure 1B). We previously reported that the slower-migrating band includes Tf carrying sialic acid terminated-glycan (Sia-Tf) [13]. The faster-migrating band includes Tf carrying *N*-acetylglucosamine terminated-glycan (GlcNAc-Tf) and mannose terminated-glycan (Man-Tf) [15]. Broad bands around the 23–24 kDa area and a sharp band around 14 kDa react with anti-L-PGDS and anti-TTR antibodies, respectively (Figure 1A,B). The migration positions of recombinant TTR and L-PGDS were shifted due to the inclusion of a His-tag motif.

### 2.2. Concentration of Major Proteins in the CSF

We previously demonstrated that CSF Man-Tf levels are increased in MCI and AD [15]. Therefore, Man-Tf, together with L-PGDS, TTR, and Tf isoforms in the CSF, was quantified for CN, MCI, and AD subjects (Figure 2) whose MMSE scores (mean ± S.D.) were 29.0 ± 1.6, 26.9 ± 1.7, and 20.4 ± 4.2, respectively. CN subjects recruited in the present study did not show evidence of dementia at baseline. However, this did not necessarily mean that CN subjects were free from AD pathology; only 28% of CN subjects had CSF tau and p-tau values that were lower than cutoff values. Indeed, five out of 23 CN subjects converted to MCI during the follow-up period (Figure 2, red filled circle). Disease controls included patients with neurodegenerative diseases such as progressive supranuclear palsy (PSP), frontotemporal degeneration (FTD), Parkinson’s disease (PD), and dementia with Lewy bodies (DLB).

Compared to disease controls, the Man-Tf levels were significantly higher in CN and MCI, and showed a tendency to be higher (*p* = 0.07) in AD (Figure 2A). Likewise, GlcNAc-Tf levels were significantly higher in all groups, indicating that Man-Tf and GlcNAc-Tf showed similarly incrementing profiles (Figure 2B). L-PGDS levels were higher only in AD subjects (*p* < 0.001) (Figure 2C), while TTR levels were slightly higher in MCI and lower in CN and AD compared to control, but differences were not significant (Figure 2D). Neither Sia-Tf nor total protein levels showed significant differences for any groups (Figure 2E,F). Taken together, with the exception of L-PGDS in MCI, CSF levels of brain-derived L-PGDS, Man-Tf, and GlcNAc-Tf showed a tendency to increase in the CSF of patients from all groups. 

### 2.3. Correlations between Major Proteins in the CSF

We previously demonstrated that CSF levels of L-PGDS and brain-derived Tf were concomitantly altered in SIH and iNPH [8]. Therefore, we analyzed Man-Tf, GlcNAc-Tf, and L-PGDS levels in the CSF samples of all subjects included in the present study; i.e., AD, MCI, CN, PSP, FTD, PD, and DLB. We found that Man-Tf levels correlated well with GlcNAc-Tf levels (*r* = 0.80) and moderately with L-PGDS levels (*r* = 0.56) (Figure 3A,B), while GlcNAc-Tf levels correlated well with L-PGDS (*r* = 0.70) (Figure 3C). In contrast, Man-Tf levels did not correlate with Sia-Tf (*r* = 0.02), TTR (*r* = 0.03) or total protein levels (*r* = 0.09) (Figure 3D–F). This result suggests that the levels of brain-derived proteins correlate well with each other regardless of the disease involved.

### 2.4. Correlations between Major CSF Proteins in CSF of CN, MCI, and AD

Correlations of the CSF proteins were also analyzed for the CN, MCI, and AD clinical groups. In CN, Man-Tf levels correlated well with GlcNAc-Tf (*rs* = 0.85) and L-PGDS (*rs* = 0.61) levels (Table 1), while GlcNAc-Tf also correlated well with L-PGDS (*rs* = 0.72). In contrast, Man-Tf levels did not correlate with Sia-Tf or TTR levels. High correlations were thus found between Man-Tf, GlcNAc-Tf, and L-PGDS in the CN group. In MCI, Man-Tf levels correlated well with GlcNAc-Tf (*rs* = 0.71) and L-PGDS (*rs* = 0.64) levels, while GlcNAc-Tf correlated well with L-PGDS (*rs* = 0.68). Man-Tf levels were negatively, but weakly, correlated with TTR (*rs* = −0.36) levels, but not with Sia-Tf. Again, high correlations were observed among brain-derived proteins in the MCI group. In AD, Man-Tf levels correlated well with GlcNAc-Tf (*rs* = 0.78) and L-PGDS (*rs* = 0.64) levels, but not with Sia-Tf or TTR. On the other hand, GlcNAc-Tf levels correlated well with L-PGDS (*rs* = 0.69) levels. These results suggest that the brain-derived proteins showed high correlations with each other across all clinical groups.

### 2.5. Correlations between Major CSF Proteins and AD Biomarkers

Our previous study demonstrated that Man-Tf and p-tau were co-expressed in hippocampal neurons in the AD brain, and that CSF Man-Tf levels correlated well with p-tau levels in MCI and AD. Other CSF proteins, such as Man-Tf, GlcNAc-Tf, L-PGDS, TTR, and Sia-Tf were therefore examined for their correlation with AD markers (Table 2). In CN subjects, the CSF proteins showed significant correlations with AD markers as follows: Man-Tf vs. p-tau (*rs* = 0.53), tau (*rs* = 0.56), Aβ40 (*rs* = 0.73), and Aβ42 (*rs* = 0.52); GlcNAc-Tf vs. p-tau (*rs* = 0.77), tau (*rs* = 0.79), and Aβ40 (*rs* = 0.76); L-PGDS vs. p-tau (*rs* = 0.91), tau (*rs* = 0.87), and Aβ40 (*rs* = 0.76). L-PGDS showed a moderate (*rs* = 0.43), but not significant, correlation with Aβ42. The Aβ42/Aβ40 ratio showed negative correlations with Man-Tf (*rs* = −0.15), GlcNAc-Tf (*rs* = −0.34), and L-PGDS (*rs* = −0.47); however, none of these were statistically significant. In contrast, neither Sia-Tf nor TTR levels correlated with any AD markers. Taken together, the brain-derived CSF proteins correlated significantly with p-tau, tau, and Aβ40 levels in the CN group, but the blood-derived proteins did not. In MCI, CSF protein levels also correlated with AD markers as follows: Man-Tf vs. p-tau (*rs* = 0.68), tau (*rs* = 0.50), Aβ40 (*rs* = 0.44); GlcNAc-Tf vs. p-tau (*rs* = 0.61), tau (*rs* = 0.51), and Aβ40 (*rs* = 0.42); L-PGDS vs. p-tau (*rs* = 0.46) and tau (*rs* = 0.37). The Aβ42/Aβ40 ratio showed negative correlations with Man-Tf (*rs* = −0.24), GlcNAc-Tf (*rs* = −0.45), and L-PGDS (*rs* = −0.23), but none were statistically significant. Overall, in the MCI group, the brain-derived proteins correlated moderately with AD markers. In AD, p-tau showed a moderate, though significant, correlation with the CSF proteins (*rs* = 0.51~0.55), but not with other AD markers. TTR negatively correlated with p-tau (*rs* = −0.41) and tau (*rs* = −0.72), while Sia-Tf correlated with tau (*rs* = −0.41). These results suggest that AD markers tend to correlate with the brain-derived CSF proteins in the AD and MCI groups; however, with the exception of p-tau, their correlation declined concomitantly with AD progression.

## 3. Discussion

It has been established that L-PGDS is secreted from the choroid plexus and pia mater [8]. Man-Tf and GlcNAc-Tf are also biosynthesized in the brain, because these Tf isoforms are undetectable in CSF of hydranencephaly patients with congenital lack of cerebrum [15]. In the previous study, we reported that the CSF proteins derived from the brain could be a biomarker of altered CSF production. In the present study, we analyzed these brain-derived protein levels in CSF samples from CN, MCI, and AD patients as well as patients with other neurodegenerative diseases, which include other tauopathy (PSP and FTD) and synucleinopathy (DLB and PD). The levels of brain-derived proteins were well correlated with each other irrespective of the disease in question, suggesting that good correlation is a common phenotype at least in these diseases, but not specific to AD. In contrast, elevations of CSF proteins are demonstrated only in AD-related groups, suggesting that the elevations are related to AD pathophysiology, although the molecular mechanism has yet to be clarified. In addition, the elevations are correlated with AD markers. In the CN group, the brain-derived CSF protein levels correlated well with those of p-tau, tau, and Aβ40, and correlated moderately with Aβ42. Similar correlation patterns were observed for the MCI group, but their correlation coefficients tended to be lower. In the AD group, p-tau correlated with brain-derived CSF protein levels, whereas other AD markers did not, suggesting that the correlation mechanism is altered in AD progression.

Besides serving as possible markers of CSF production, the brain-derived CSF proteins have Aβ binding activity, which may potentially prevent AD progression by affecting Aβ metabolism. Raditsis et al. reported that Tf inhibited Aβ aggregation by binding to Aβ oligomers as well as monomers [16]. Another major protein, L-PGDS, was demonstrated to bind Aβ monomers and fibrils to thereby inhibit not only self-aggregation but also seed-dependent aggregation [17]. In addition, L-PGDS-deficient mice showed a 3.5-fold higher Aβ deposition than control mice after the intraventricular infusion of Aβ42 [18]. These results suggest that CSF proteins prevent the aggregation and deposition of Aβ in the brain. Further studies on potentially neuroprotective CSF proteins could lead to the development of new therapeutic strategies.

Quantifying AD markers in plasma will be promising laboratory tests for AD diagnosis [19,20]. Moscoso et al. reported that plasma p-tau181 is a marker for monitoring neurodegeneration and cognitive decline in AD [21]. Tatebe et al. reported, however, that the plasma p-tau concentration is only 0.1% of that of the CSF [19], requiring sensitive methods. A potential advantage of the brain-derived proteins is their high concentration in CSF; e.g., L-PGDS concentrations of CSF and serum are 5–15 μg/mL and 50–200 ng/mL, respectively, [22] which are measurable by a conventional ELISA kit. Indeed, Tsutsumi et al. reported that L-PGDS concentration in a skull vein is significantly higher than that of peripheral veins, suggesting that CSF L-PGDS could diffuse into a vein and provide information on L-PGDS in CSF [22]. This may be also the case with other CSF proteins, because Man-Tf and GlcNAc-Tf show high concentrations in CSF: 2–8 μg/mL for Man-Tf and 3–11 μg/mL for GlcNAc-Tf [15]. Quantifying the CSF proteins in plasma would give a clue to monitor CSF metabolism in neurological diseases.

## 4. Materials and Methods

### 4.1. Patients

Patients with AD, MCI, or other neurodegenerative disorders were consecutively recruited from Fukushima Medical University, Tohoku University, and Juntendo University. MMSE score criteria for AD, MCI, and CN were <25, 25~27 and >28, respectively. For each disease, the number of patients, age (mean ± S.D.), male/female distribution, and MMSE are listed in Table 3. Disease diagnosis was based on the following criteria: AD, “The diagnosis of dementia due to Alzheimer’s disease: Recommendations from the National Institute on Aging-Alzheimer’s Association workgroups on diagnostic guidelines for Alzheimer’s disease.” [1]; MCI, “The diagnosis of mild cognitive impairment due to Alzheimer’s disease: recommendations from the National Institute on Aging-Alzheimer’s Association workgroups on diagnostic guidelines for Alzheimer’s disease” [23]; progressive supranuclear palsy (PSP), “Accuracy of clinical criteria for the diagnosis of progressive supranuclear palsy” [24]; frontotemporal degeneration (FTD), “Frontotemporal lobar degeneration: a consensus on clinical diagnostic criteria” [25]; dementia with Lewy bodies (DLB) “Diagnosis and management of dementia with Lewy bodies: Fourth consensus report of the DLB Consortium.” [26]; Parkinson’s disease (PD), “the Queen Square Brain Bank criteria for the diagnosis of Parkinson’s disease” (https://www.mims.ie/news/queen-square-brain-bank-qsbb-criteria-for-pd-diagnosis-02-04-2013/, accessed on 14 March 2022).

### 4.2. SDS-PAGE and Blotting Analyses

CSF and serum samples were dissolved in Laemmli buffer without 2-mercaptoethanol, boiled for 3 min, and loaded on SDS-polyacrylamide gels (194–1502, FUJIFILM Wako) [27]. After SDS-PAGE, protein bands were visualized with a Silver Stain II kit (FUJIFILM Wako). For immunoblotting, proteins separated on gels were transferred to nitrocellulose membranes. The membranes were blocked in 3% skim milk, and incubated sequentially with the following combinations of primary and secondary antibodies: anti-Tf antibody (Bethyl Laboratories, Montgomery, TX, USA) and horseradish peroxidase (HRP)-labeled anti-goat IgG (Jackson ImmunoResearch Laboratories, West Grove, PA, USA); anti-PDGS antibody (PA1-46023, Thermo Fisher Scientific, Waltham, MA, USA) and HRP-labeled anti-rabbit IgG (Jackson ImmunoResearch Laboratories, West Grove, PA, USA); anti-TTR antibody (ab9015, Abcam, Cambridge, CB2 0AX, UK) and HRP-labeled anti-sheep IgG antibody (#31480, BiotechnologyThermo Fisher Scientific). The protein bands were visualized with a SuperSignal West Dura Chemiluminescence Substrate Kit (Pierce Biotechnology, Rockford, IL, USA). Purified standards of GlcNAc and Man-Tf mixture were purified from CSF as described previously [15]. His-tagged TTR (89-7754-27) and L-PGDS (E-PKSH030653.10) were purchased from Elabscience Biotechnology Inc., (Houston, TX, USA) and Biomol (Hamburg, Germany)

### 4.3. ELISA for CSF Proteins

CSF samples were aliquoted and stored at −80 °C until use. Repeated freeze-thawing (>two times) was avoided. Each assay was performed in triplicate. For quantifying Man-Tf, a lectin-ELISA was developed according to Shirotani et al. [27] with slight modification. Briefly, a 96-well plate (C8 Maxisorp Nunc-Immuno Module plate, Nunc, Roskilde, Denmark) was coated with rabbit anti-Tf antibody (1 μg/mL) (A0061, Dako Ltd./Agilent Technologies, Inc., Santa Clara, CA, USA) in 100 mM carbonate buffer at 4 °C overnight. Plates were washed with Tris buffered saline (TBS) and then blocked at room temperature for 1 h with 10% N101 (S410-0301, NOF Corp., Tokyo, Japan) in TBS. They were then washed with TBS containing 0.05% Tween 20 (#1706531, Bio-Rad Laboratories, Inc., Hercules, CA, USA) (TBST). The CSF samples were pre-treated at 55 ℃ for 1 h in the presence of 10~20 μL of phosphate buffered saline (PBS) containing 0.6% 2-mercaptoethanol (#1610710, Bio-Rad Laboratories, Inc.) and 0.003% SDS (S0588, Tokyo Chemical Industry Co., Ltd., Tokyo, Japan). The sample solution was appropriately diluted with TBST, applied to plates, and incubated overnight at 4 °C. The plates were then washed three times with TBST. The target molecule, Man-Tf, captured on each plate was incubated with biotinylated rBC2L-A lectin (1 μg/mL) in TBST containing 10 mM CaCl_2_ (TBST-CaCl_2_) at room temperature for 2 h. Plates were washed twice with TBST-CaCl_2_, and horseradish peroxidase-labeled streptavidin (50 ng/mL) (N100, Thermo Fisher Scientific) in TBST-CaCl_2_ was added and incubated for 2 h at room temperature, following which plates were washed twice with TBST-CaCl_2_ and TMB Micro well Peroxidase Substrate System (#50-76-11, Kirkegaard and Perry Laboratories, Inc., Gaithersburg, MD, USA) then added. Color development was stopped by adding 1N phosphoric acid. Absorbances were measured at 450 nm by a Varioskan LUX multimode microplate reader (Thermo Fisher Scientific). Assays were performed in triplicate. For quantifying Sia-Tf and GlcNAc-Tf, the lectin probes used were SSA (197-10371, FUJIFILM Wako) and PVL (Medical and Biological Laboratories Co., Ltd., Nagoya, Japan), respectively [27]. Total Tf was quantified with a Human Transferrin ELISA Quantitation Set (E80-128-23, Bethyl Laboratories). AD core markers were assayed by LSI Medience Corporation (Tokyo, Japan), using the following ELISA kits: Phinoscholar hTAU (10-992, Nipro Parma Corporation, Osaka, Japan) for (total) tau; Phinoscholar pTAU (10-994, Nipro) for p-tau (181); Human βAmyloid (1-40) ELISA Kit Wako II (298-64601, FUJIFILM Wako) for Aβ40; Human βAmyloid (42) ELISA Kit Wako High Sensitive (292-64501, FUJIFILM Wako) for Aβ42.

### 4.4. Statistical Analyses

Statistical analyses were performed using SPSS software (version 26). Data normality was examined by the Shapiro–Wilk test. Significant differences among multiple comparisons were assessed by the Kruskal–Wallis method followed by Bonferroni correction.

## Figures and Tables

**Figure 1 metabolites-12-00355-f001:**
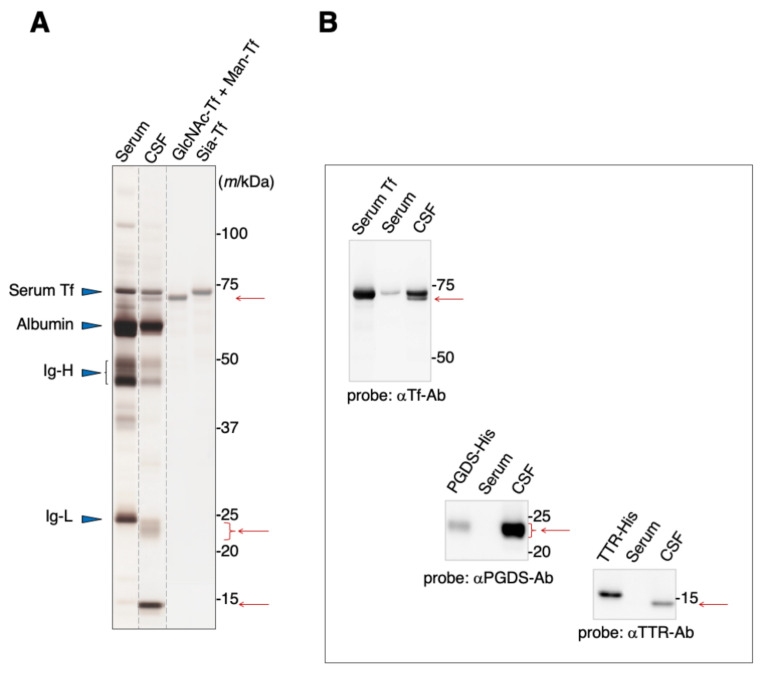
SDS-PAGE analysis of human serum and CSF (**A**). Protein bands were visualized by silver staining. Migration positions of major serum proteins such as serum Tf, albumin, and immunoglobulin heavy chain (Ig-H) and light chain (Ig-L) are indicated by blue triangles. Positions of additional major proteins in CSF are indicated by red arrows. Each lane separated on the gel is indicated by dotted lines. Purified standards of GlcNAc-Tf and Man-Tf were obtained from CSF as previously described, reprinted with permission from ref. [15], 2021 [15], Yasuhiro Hashimoto. The presence of TTR and L-PGDS, in addition to Man-Tf and GlcNAc-Tf, in the CSF was confirmed by immunoblotting using specific antibodies (**B**). Recombinant His-tagged TTR and L-PGDS show a mobility shift.

**Figure 2 metabolites-12-00355-f002:**
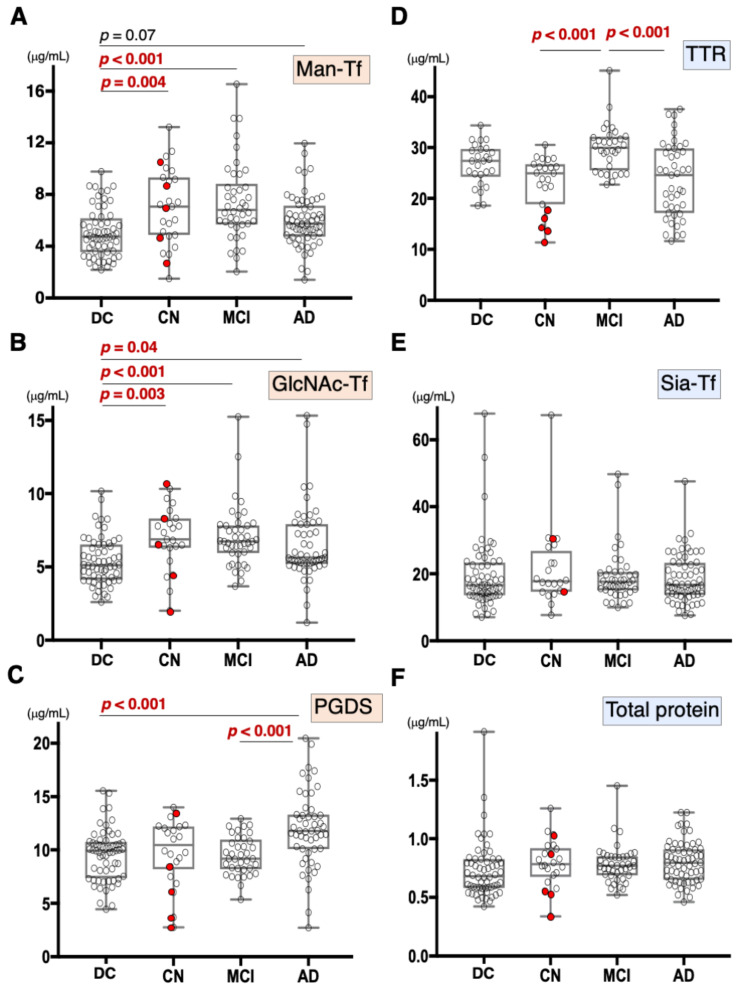
Box plots of CSF marker concentrations in CN, MCI, AD, and disease control (DC), which includes FTD, PSP, DLB, and PD: Man-Tf (**A**), GlcNAc-Tf (**B**), L-PGDS (**C**), TTR (**D**), Sia-Tf (**E**), and total protein (**F**). Horizontal lines within boxes show median values; boxes exclude upper and lower interquartile ranges; whiskers indicate the maximum and the minimum values. Five CN subjects who converted to MCI are indicated with red filled circles. Sia-Tf data from three conversion cases were not determined. Man-Tf data of disease control, MCI, AD is cited from our previous paper, reprinted with permission from ref. [15], 2021, Yasuhiro Hashimoto. Multiple comparisons were assessed by the Kruskal–Wallis method followed by Bonferroni correction. All combinations of clinical groups were assessed, and combinations showing significant difference were indicated with horizontal bars except that Man-Tf data between DC and AD (*p* = 0.07).

**Figure 3 metabolites-12-00355-f003:**
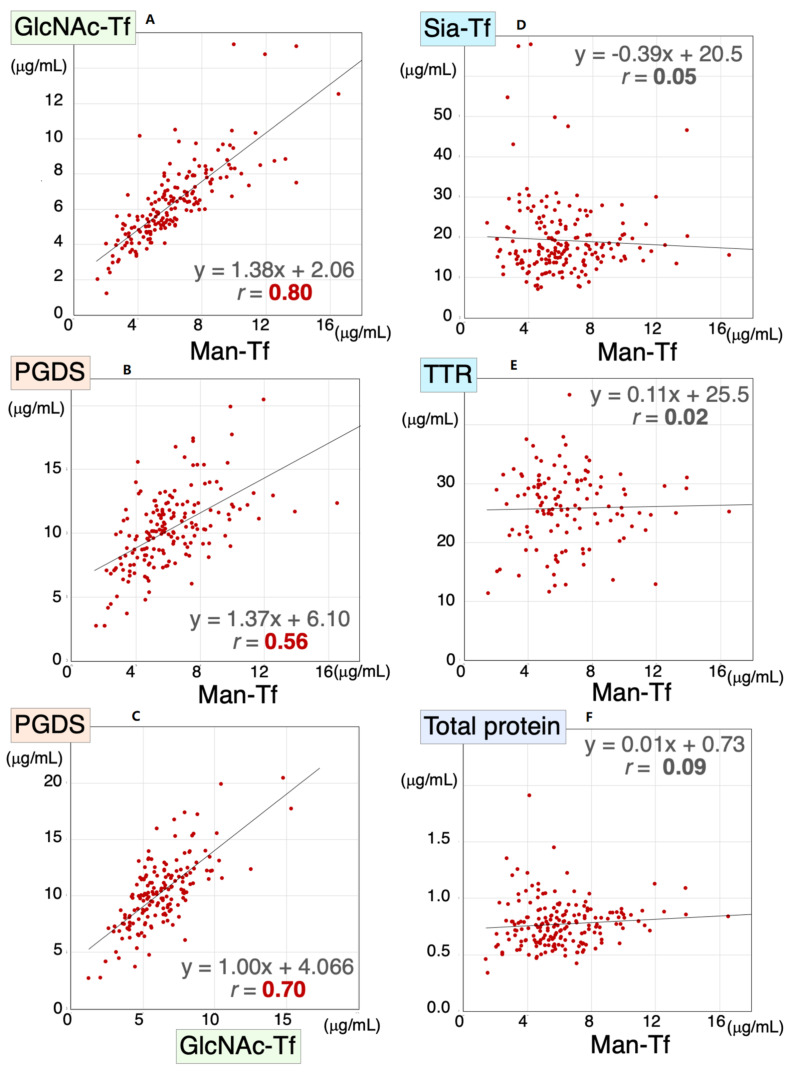
Scatter diagrams for Man-Tf and other CSF proteins for all subjects included in the present study. Correlation coefficients and regression line equations for Man-Tf versus GlcNAc-Tf, L-PGDS, TTR, Sia-Tf and total protein are shown. Data for TTR levels in PD were not determined. Man-Tf data, except for CN subjects, are cited from our previous paper, reprinted with permission from ref. [15], 2021, Yasuhiro Hashimoto.

**Table 1 metabolites-12-00355-t001:** Correlation coefficients among CSF major proteins in CN, MCI and AD.

CN		Man-Tf	GlcNAc-Tf	L-PGDS	TTR	Sia-Tf
**Man-Tf**	*rs*	1.00	**0.85 *****	**0.61 ****	−0.02	−0.34
**GlcNAc-Tf**	*rs*	**0.85 *****	1.00	**0.72 *****	0.05	0.02
**L-PGDS**	*rs*	**0.61 ****	**0.72 *****	1.00	0.37	0.21
**TTR**	*rs*	−0.02	0.05	0.37	1.00	0.37
**Sia-Tf**	*rs*	−0.34	0.02	0.21	0.37	1.00
**MCI**		**Man-Tf**	**GlcNAc-Tf**	**L-PGDS**	**TTR**	**Sia-Tf**
**Man-Tf**	*rs*	1.00	**0.71 *****	**0.64 *****	−0.36	0.01
**GlcNAc-Tf**	*rs*	**0.71 *****	1.00	**0.68 *****	−0.29	0.17
**L-PGDS**	*rs*	**0.64 *****	**0.68 *****	1.00	−0.11	0.33
**TTR**	*rs*	−0.36 *	−0.29	−0.11	1.00	0.19
**Sia-Tf**	*rs*	0.01	0.17	0.33	0.19	1.00
**AD**		**Man-Tf**	**GlcNAc-Tf**	**L-PGDS**	**TTR**	**Sia-Tf**
**Man-Tf**	*rs*	1.00	**0.78 *****	**0.64 *****	−0.07	0.02
**GlcNAc-Tf**	*rs*	**0.78 *****	1.00	**0.69 *****	0.01	0.14
**L-PGDS**	*rs*	**0.64 *****	**0.69 *****	1.00	−0.13	0.37 *
**TTR**	*rs*	−0.07	0.01	−0.13	1.00	0.40 *
**Sia-Tf**	*rs*	0.02	0.14	0.37 *	0.40 *	1.00

Spearman’s rank correlation coefficients (*rs*) with values higher than 0.50 are highlighted with bold letters in yellow cells. Man-Tf data of MCI and AD are cited from our previous paper [15]. Probability levels: * *p* < 0.05, ** *p* < 0.01, *** *p* < 0.001.

**Table 2 metabolites-12-00355-t002:** Correlation coefficients among CSF major proteins and AD biomarkers in CN, MCI and AD.

CN		p-tau	tau	Aβ40	Aβ42	Aβ42/Aβ40
**Man-Tf**	*rs*	**0.53 ***	**0.56 ***	**0.73 ****	**0.52 ***	−0.15
**GlcNAc-Tf**	*rs*	**0.77 *****	**0.79 *****	**0.76 *****	0.49 *	−0.34
**L-PGDS**	*rs*	**0.91 *****	**0.87 *****	**0.76 *****	0.43	−0.47
**TTR**	*rs*	0.12	0.14	0.01	0.08	0.02
**Sia-Tf**	*rs*	0.37	0.34	0.13	−0.03	−0.27
**MCI**		**p-tau**	**tau**	**Aβ40**	**Aβ42**	**Aβ42/Aβ40**
**Man-Tf**	*rs*	**0.68 *****	**0.50 ****	0.44 *	0.41	−0.24
**GlcNAc-Tf**	*rs*	**0.61 *****	**0.51 ****	0.42 *	0.40	−0.45
**L-PGDS**	*rs*	0.46*	0.37 *	0.31	0.40	−0.23
**TTR**	*rs*	−0.19	−0.15	−0.33	−0.40	0.12
**Sia-Tf**	*rs*	0.18	0.01	0.38 *	0.46	−0.15
**AD**		**p-tau**	**tau**	**Aβ40**	**Aβ42**	**Aβ42/Aβ40**
**Man-Tf**	*rs*	**0.55 ****	0.17	−0.02	0.00	0.00
**GlcNAc-Tf**	*rs*	**0.51 ****	0.25	−0.08	0.32	0.21
**L-PGDS**	*rs*	**0.53 ****	0.18	0.05	0.39	0.32
**TTR**	*rs*	−0.41 *	−0.72 ***	−0.09	−0.07	−0.32
**Sia-Tf**	*rs*	−0.17	−0.41 *	0.25	0.54	0.25

Spearman’s rank correlation coefficients (*rs*) with values higher than 0.50 are highlighted with bold letters in yellow cells. Man-Tf data of MCI and AD are cited from our previous paper [15]. Probability levels: * *p* < 0.05, ** *p* < 0.01, *** *p* < 0.001.

**Table 3 metabolites-12-00355-t003:** Patient Characteristics.

Disease	Age * (years)	Patient Number	Gender (M/F)	MMSE *
AD: Alzheimer’s disease	73.5 ± 8.7	61	27/34	20.4 ± 4.2
MCI: mild cognitive impairment	76.0 ± 6.9	42	19/23	26.9 ± 1.7
CN: cognitively normal	72.3 ± 9.0	23	10/13	29.0 ± 1.6
PSP: progressive supranuclear palsy	69.6 ± 5.7	7	4/3	n.d.
FTD: frontotemporal degeneration	60.7 ± 4.1	10	5/5	n.d.
DLB: dementia with Lewy bodies	66.8 ± 2.6	9	6/3	n.d.
PD: Parkinson’s disease	68.2 ± 8.5	34	13/21	n.d.

* mean ± S.D.; n.d., not determined.

## Data Availability

All data are contained within the article.

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
