# Peer review of "High Correlation among Brain-Derived Major Protein Levels in Cerebrospinal Fluid: Implication for Amyloid-Beta and Tau Protein Changes in Alzheimer’s Disease"

_metabolites, 2022, doi:10.3390/metabo12040355_

Round 1
Reviewer 1 Report
The authors should implement the introduction by adding more studies and bibliographical notes.
Author Response
Comment:
The authors should implement the introduction by adding more studies and bibliographical notes.
Response to the comment:
Thank you for the comment. We assume that the reviewer ask us to describe glymphatic system or studies on CSF flow in detail. Another reviewer also mentioned, “the authors did not cite the entire spectrum of the controversial experimental literature that is existing, but instead they rely on just one work they cite under (8). The present results do not imply any correlation with CSF turnover. Also, the subject of CSF turnover in disease and CSF clearance of brain metabolites and “clearance” is still a matter of great debate and controversial and lacking experiments.” Finally, the reviewer suggested that the authors remove those speculative contents and simply remain descriptive.” We decided to follow this suggestion. Namely, we remove speculative contents and simply remain descriptive. The terms of “clearance,” “turnover,” are removed. Especially, we removed sentences about glymphatic system; e.g., “CSF turnover facilitates the clearance of metabolic waste and helps maintain optimal solute concentrations for neuronal function [8]. Jessen et al. hypothesized that the “glymphatic system” plays a major role in generating CSF turnover; i.e., body fluid in para-arterial spaces enters the brain parenchyma, collects waste, and returns to the lymphatic system via para-venous spaces [17, 18].The glymphatic system failure may be partly responsible for accumulating extracellular Aβ and tau proteins, although its involvement in AD has been yet to be fully clarified [19].” (in line 333-339 of previous manuscript). We also removed sentences related to previous reference No.8, 17, 18, 19.
Nevertheless, some reader may question why CSF proteins are highly correlated each other. As a possible explanation, we would like to hypothesize that CSF proteins are markers of altered CSF production, which is based on our previous studies on SIH and iNPH. The hypothesis is not fully proved, but could be a possible explanation.
In conclusion, we eliminate or minimize speculative description including glymphatic system. Instead, we inserted sentences for describing our previous studies on SIH and iNPH in detail in the second paragraph of introduction section (line 117-134).
Reviewer 2 Report
This study is important to support CSF biomarkers in the diagnosis of dementia and cognitive impairment. The study methods are robust, and the study is well conducted. Authors propose that “In the present study we analyzed CSF levels of brain-derived Tf, L-PGDS, and AD markers in neurodegenerative diseases and examined their correlation with possible “CSF turnover.”, and the authors conclude “The present study revealed that levels of brain-derived proteins were well correlated with each other irrespective of the disease in question, suggesting that these proteins could be markers of CSF turnover not only in SIH/iNPH but also in other neurodegenerative diseases. Especially, high correlations between these proteins were detected in CN, MCI, and AD groups. Another conspicuous finding here is that the brain-derived proteins showed a tendency to increase in all clinical groups. One possible and attractive explanation of this finding is that the brain-derived CSF proteins may reflect altered CSF turnover and waste clearance in neurological diseases including AD”.
The aims and conclusions are not supported by the study design. Figure 2 shows the significant overlap of brain derived CSF biomarker values/levels between all the groups (so no meaningful differences noticed) , and as such, the results shown in Figure 3 are mere statistical correlations and the true pathophysiological implications is truly questioned. It is suggested that the authors remove those speculative contents and simply remain descriptive. This is valuable enough to the scientific community.
The authors should simply state that “In the present study we analyzed CSF levels of brain-derived Tf, L-PGDS, and AD markers in neurodegenerative diseases” and not imply any correlation with CSF turnover. Also, the submect of CSF turnover in disease and CSF clearance of brain metabolites and “clearance” is still a matter of great debate and controversial and lacking experiments. The authors have not cited the entire spectrum of the controversial experimental literature that is existing, but instead they rely on just one work they cite under (8). It is suggested to the authors to rather concentrate on the diagnostic value of their study and the simple description of their findings as proposed above.
As a last critique, it is not clear what the control diseases (PD, LBD etc.) , other than in figure 2, really contribute to the study. The authors should keep those in , but should comment on this in the discussion.
Author Response
Comment
This study is important to support CSF biomarkers in the diagnosis of dementia and cognitive impairment. The study methods are robust, and the study is well conducted. Authors propose that “In the present study we analyzed CSF levels of brain-derived Tf, L-PGDS, and AD markers in neurodegenerative diseases and examined their correlation with possible “CSF turnover.”, and the authors conclude “The present study revealed that levels of brain-derived proteins were well correlated with each other irrespective of the disease in question, suggesting that these proteins could be markers of CSF turnover not only in SIH/iNPH but also in other neurodegenerative diseases. Especially, high correlations between these proteins were detected in CN, MCI, and AD groups. Another conspicuous finding here is that the brain-derived proteins showed a tendency to increase in all clinical groups. One possible and attractive explanation of this finding is that the brain-derived CSF proteins may reflect altered CSF turnover and waste clearance in neurological diseases including AD”.
The aims and conclusions are not supported by the study design (Comment #1). Figure 2 shows the significant overlap of brain derived CSF biomarker values/levels between all the groups (so no meaningful differences noticed) (Comment #2), and as such, the results shown in Figure 3 are mere statistical correlations and the true pathophysiological implications is truly questioned. It is suggested that the authors remove those speculative contents and simply remain descriptive. This is valuable enough to the scientific community (Comment #3). The authors should simply state that “In the present study we analyzed CSF levels of brain-derived Tf, L-PGDS, and AD markers in neurodegenerative diseases” and not imply any correlation with CSF turnover. Also, the subject of CSF turnover in disease and CSF clearance of brain metabolites and “clearance” is still a matter of great debate and controversial and lacking experiments. The authors have not cited the entire spectrum of the controversial experimental literature that is existing, but instead they rely on just one work they cite under (8). It is suggested to the authors to rather concentrate on the diagnostic value of their study (Comment #4) and the simple description of their findings as proposed above (Comment #3).
We divided the comment into Comment #1-4, which are indicated in parenthesis. Responses to the comment are as follows.
Response to the comment #1, 3
Thank you for the suggestion, “the authors remove those speculative contents and simply remain descriptive. This is valuable enough to the scientific community.” According to the suggestion we modified all over the text to concentrate on simply describing data, and remove or minimize speculative contents. In addition, the terms of “clearance”, “flow” and “turnover” are eliminated. Especially, we removed sentences about glymphatic system; e.g., “CSF turnover facilitates the clearance of metabolic waste and helps maintain optimal solute concentrations for neuronal function [8]. Jessen et al. hypothesized that the “glymphatic system” plays a major role in generating CSF turnover; i.e., body fluid in para-arterial spaces enters the brain parenchyma, collects waste, and returns to the lymphatic system via para-venous spaces [17, 18].The glymphatic system failure may be partly responsible for accumulating extracellular Aβ and tau proteins, although its involvement in AD has been yet to be fully clarified [19].” (in line 333-339 of previous manuscript). We also removed sentences related to previous reference No.8, 17, 18, 19.
Nevertheless, some reader may question why CSF proteins are highly correlated each other. As a possible explanation, we would like to hypothesize that CSF proteins are markers of altered CSF production, which is based on our previous studies on SIH and iNPH. The hypothesis is not fully proved, but could be a possible explanation. Accordingly, we would like to insert sentences for describing our previous studies on SIH and iNPH in detail in the second paragraph of introduction section (line 117-134).
Response to Comment #2, 4:
I agree with the reviewer that the brain-derived CSF proteins show significant overlap, suggesting that their diagnostic accuracies are not good. We tentatively examine the sensitivity and specificity of GlcNAc-Tf for diagnosing AD-related groups versus control diseases. For AD, sensitivity and specificity are 77%, 53%, respectively. For MCI, sensitivity and specificity are 79%, 67%, respectively. For CN, sensitivity and specificity are 78%, 72%, respectively. Thus, the diagnostic accuracies are not good enough for clinical use. Nevertheless, GlcNAc-Tf increases significantly in AD-related groups but not control diseases, which would be worth describing as the reviewer suggested.
Comment #5:
As a last critique, it is not clear what the control diseases (PD, LBD etc.), other than in figure 2, really contribute to the study. The authors should keep those in, but should comment on this in the discussion.
Response to comment #5:
Our interpretation is the following. Correlations of CSF proteins are detected not only in AD-related groups but also the control diseases, which include other tauopathy (PSP and FTD) and synucleinopathy (DLB and PD), indicating that the good correlation is a common phenotype at least in these diseases, but not specific to AD. In contrast, elevations of CSF proteins are demonstrated only in AD-related groups, suggesting that the elevations are related to AD pathophysiology although the molecular mechanism have yet to be clarified. In addition, the elevations are correlated with AD markers in CN subjects whereas the correlations decline in MCI and AD subjects, suggesting that the correlation mechanism is altered in AD progression.
According to the reviewer’s suggestion, we inserted the interpretation in discussion section (line 366-372) as follows: “The levels of brain-derived proteins were well correlated with each other irrespective of the disease in question, suggesting that the good correlation is a common phenotype at least in these diseases, but not specific to AD. In contrast, elevations of CSF proteins are demonstrated only in AD-related groups, suggesting that the elevations are related to AD pathophysiology although the molecular mechanism has yet to be clarified. In addition, the elevations are correlated with AD markers. - - -”
Reviewer 3 Report
The paper is very good. There are no observable faults. However, due to the abundance of acronyms and the comparisons between groups and scruples, the reedibility is poor, and maybe a series of tables would have been better.
Author Response
Comment
The paper is very good. There are no observable faults. However, due to the abundance of acronyms and the comparisons between groups and scruples, the readibility is poor, and maybe a series of tables would have been better.
Response to the comment
I agree with the reviewer that the readability is not good. However, disease names need to be abbreviated due to their long names; progressive supranuclear palsy (PSP), idiopathic normal pressure hydrocephalus (iNPH), etc. Use of these abbreviations for diseases and AD marker names are popular for this kind of paper. Unpopular abbreviations are for transferrin isoforms. Isoforms have the same protein and only glycan portions are different. Therefore, abbreviations including name of terminal sugar would be reasonable; Sia-Tf for sialic acid-terminated transferrin and Man-Tf for mannose-terminated transferrin, etc.
Using a series of tables may be a good idea, but we would like to publish the present form.
Reviewer 4 Report
Understanding biochemical changes in CSF is critical in identifying the fluid biomarkers for aging diseases such as Alzheimer’s’ diseases. The study by Kyoka Hoshi et al. attempted to find the correlation among the CSF proteins that are derived from brain among AD patients (stages of AD patients). This is intriguing study and provides the information on proteins that are not conventional probed in CSF fluid biomarkers for Alzheimer’s’ diseases. However, there are few minor corrections to be made before the final acceptance of the manuscript
Major comments
- In line 41 and 42 in Abstract, the values for MMSE for AD, MCI, and CN are wrongly mentioned. It should be corrected (Refer table 3)
- In line 128 in Results 2.2 section, the values for MMSE for AD, MCI, and CN are wrongly mentioned. It should be corrected (Refer table 3)
- Figure 2, the multiple comparison should be done between all inter groups. Based on the p value description, I see it is done only with the reference to the diseased controls. To appreciate if there are any changes among the inter groups (for example CN vs MCI, or MCI vs AD, CN vs AD) they should be evaluated.
- In Figure 2, controls should be replaced as diseased controls in the Figure (Maybe abbreviation such as DC- diseased control). Controls in Figure 2 can be misinterpreted to cognitively normal healthy people, which is CN in your study.
Minor comments
- In all the figures there is an additional wording about figure number at the top right corner. This should be deleted.
- Sometimes it is L-PGDS and sometimes it is PGDS. Mention consistently.
- In Line 332 typo error. Change “purifiedfrom” to “purified from”. Look out for other such typo throughout the Manuscript.
Author Response
Major comments
Comment #1: In line 41 and 42 in Abstract, the values for MMSE for AD, MCI, and CN are wrongly mentioned. It should be corrected (Refer table 3)
Comment #2: In line 128 in Results 2.2 section, the values for MMSE for AD, MCI, and CN are wrongly mentioned. It should be corrected (Refer table 3)
Response to comment #1 and #2:
Thank you for the comments. We corrected MMSE scores in summary and results section.
Comment #3: Figure 2, the multiple comparison should be done between all inter groups. Based on the p value description, I see it is done only with the reference to the diseased controls. To appreciate if there are any changes among the inter groups (for example CN vs MCI, or MCI vs AD, CN vs AD) they should be evaluated.
Response to comment #3:
According to reviewer’s suggestion, the multiple comparison was done between all intergroups. The results were inserted in figure 2 and figure legend was modified as follows. “All combinations of clinical groups were assessed, and combinations showing significant difference were indicated with horizontal bars except that Man-Tf data between disease control and AD is not statistically significant (p = 0.07)”
Comment #4: In Figure 2, controls should be replaced as diseased controls in the Figure (Maybe abbreviation such as DC- diseased control). Controls in Figure 2 can be misinterpreted to cognitively normal healthy people, which is CN in your study.
Response to comment #4:
According to the suggestion, the terms of “controls” were replaced with DC.
Minor comments
Comment #5: In all the figures there is an additional wording about figure number at the top right corner. This should be deleted.
Response to comment #5: We deleted figure number.
Comment #6: Sometimes it is L-PGDS and sometimes it is PGDS. Mention consistently.
Response to comment #6: We replace the term of “PGDS” with “L-PGDS”
Comment #7: In Line 332 type error. Change “purifiedfrom” to “purified from”. Look out for other such typo throughout the Manuscript.
Response to comment #7: We corrected the type error.